# Calibration of Industrial Cameras for Aerial Photogrammetric Mapping

**Jakub Kolecki [1], Przemysław Kuras [1] , Elżbieta Pastucha [2] , Krystian Pyka [1],\* and Maciej Sierka [3]**

[1] Faculty of Mining Surveying and Environmental Engineering, AGH University of Science and Technology, al. Mickiewicza 30, 30-059 Kraków, Poland; kolecki@agh.edu.pl (J.K.); kuras@agh.edu.pl (P.K.)

[2] The Maersk Mc-Kinney Moller Institute, University of Southern Denmark, Campusvej 55, 5230 Odense, Denmark; elpa@mmmi.sdu.dk

[3] MGGP Aero Sp. z o.o., ul. Kaczkowskiego 6, 33-100 Tarnów, Poland; msierka@mggpaero.com

**\*** Correspondence: krisfoto@agh.edu.pl

**Abstract:** This paper details the development of a camera calibration method purpose-built for use in photogrammetric survey production. The calibration test field was established in a hangar, where marker coordinates were measured using a high-precision survey methodology guaranteeing very high accuracy. An analytical model for bundle adjustment was developed that does not directly use the coordinates of field calibration markers but integrates bundle adjustment and survey observations into a single process. This solution, as well as a classical calibration method, were implemented in a custom software, for which the C++ source code repository is provided. The method was tested using three industrial cameras. The comparison was drawn towards a baseline method, OpenCV implementation. The results point to the advantages of using the proposed approach utilizing extended bundle adjustment.

**Keywords:** camera calibration; bundle adjustment; test field; precise survey

## 1. Introduction

### 1.1. Motivation

Precise camera calibration is indispensable to photogrammetric calculations, where the accuracy of these calculations translates directly into that of the end product [1]. It is the process of determining the parameters needed for the precise reconstruction of the bundle geometry within a camera.

The aerial photogrammetric mapping (APM) of topographic features, usually applying to areas over 100 km$^2$, has utilized specialized cameras (called metric or large format cameras) for years. Typical analogue photogrammetric cameras use a $23 \times 23$ cm$^2$ frame. Contemporary metric cameras have a smaller frame, but their resolution reaches up to 400 megapixels. The end image is a computed combination of panchromatic and RGB images, where each lens has minimal distortion and is precisely calibrated within the whole lens system. Large format cameras are also equipped with motion blur compensation and are set in a rigid body that should ensure high stability of each element. The large format camera calibrations are updated every couple of years by the camera's manufacturers. Two major drawbacks of those cameras are the size (several dozen kilograms) and price (several times the price of the highest quality industrial cameras).

Over 10 years ago, smaller, cheaper, industrial-grade cameras were introduced into photogrammetry. Particularly interesting for photogrammetric survey production are the so-called medium format cameras with a minimum resolution of 40 megapixels [2]. They have one lens

equipped with a central leaf shutter (global shutter in simpler models) and can be joined into synchronized multi-camera systems. Today, the growing market of medium format cameras and other industrial cameras offers products with continually increasing resolutions and capabilities that provide a fine tradeoff between the quality of data and cost. With small dimensions, reasonably designed mechanical and communication interfaces, and easy lens replacement, medium format cameras provide considerable flexibility in use as they can, without great effort, be moved from one system to another and used onboard various kinds of aircraft, like gyrocopters and ultralight planes, including, for the lightest models, unmanned aerial vehicles (UAVs). However, this versatility demands more frequent camera calibration to maintain sufficient geometric quality of the products. Moreover, the number of photogrammetric projects relies only on the geo-referencing images obtained directly from global navigation satellite systems (GNSSs) and inertial navigation system (INS), and involves no ground control information, leaving little space to compensate for inaccuracies in the parameters of the camera during bundle adjustment. In such circumstances, camera calibration becomes an important issue that requires careful treatment.

Appropriate geometric camera calibration requires an appropriate calibration test field to minimize the influence of unstable parameters. Theoretically, the best calibration test field consists of multiple permanent ground control points spread throughout a large area. A series of aerial images are then taken directly after the aircraft takes off and just before it heads to the photogrammetric mission area to minimize the impact of parameter instability that is caused, among other factors, by changing environmental conditions (i.e., thermal) [3]. However, such an approach can be cumbersome as well as costly. As an alternative, indoor calibration is much easier to complete independent of weather conditions. The most commonly used chessboard-style printed calibration test fields are often used in robotics, and allow for the fast and convenient calibration of small industrial cameras. However, their application to calibrating medium format cameras is unfeasible because they are too small, and do not allow for the long imaging distances required to obtain sharp images with a nearly infinitely focused lens—a typical usage scenario in aerial photogrammetry surveys. Alternatively, interior, large-scale calibration test fields of targets measured with high precision can provide a reasonable tradeoff between size and calibration capabilities.

A camera calibration methodology presented in this paper was created for MGGP Aero company photogrammetric production. Two conditions were set:

- The calibration test field should be placed in the company's hangar while not limiting its working area; and
- Reprojection error (RE) of the calibration should not exceed 0.3 pixel.

To meet these conditions, we decided to place several hundred markers on the hangar wall. Although such a calibration test field resembles a 2-D solution, it cannot be treated as such and its control point coordinates have to be established precisely in 3-D space. To estimate the interior orientation parameters of calibrated cameras, the bundle adjustment method, based on collinearity equations, was utilized. The commonly implemented method was extended with the inclusion of equations describing survey measurements.

The remainder of this paper is structured as follows: In Section 1.2, the state of the art in research on the calibration of pinhole cameras is briefly presented, and Section 2 gives an overview of the presented method. Section 3 describes the preparation of the calibration test field. Section 4 presents mathematical models implemented in the solution. Section 5 discusses the results of test camera calibrations, and Section 6 gives the conclusions of this paper.

*1.2. State of the Art in Camera Calibration*

The calibration of stable pinhole cameras is based on a model formulated by Brown [4]. The classic Brown model includes the focal length and the position of the principal point, known as the internal orientation (IO), as well as three coefficients of radial distortion and two of decentering distortion.

These are immanent features of optical lenses and are known as additional parameters (AP) [1]. Brown's model is sometimes extended by using parameters that express the lack of flatness of the plane, in-plane image distortion, or other distortions [5]. Even though many studies have addressed different sets of APs, the eight-parameter set is the optimal solution for camera calibration and has become the accepted standard [1,6]. Sometimes additional affinity and shear components are added to the model, though nowadays production and consumer-grade cameras rarely exhibit those deformations [7]. The well-known collinearity equations, which contain the eight-parameter set of APs, are shown in Equation (1):

$$
\begin{aligned}
x &= x_0 - f\,\tfrac{\acute{x}}{\acute{z}} + x\big(k_1 r^2 + k_2 r^4 + k_3 r^6\big) + p_1\big(r^2 + 2x^2\big) + 2p_2 xy \\
y &= y_0 - f\,\tfrac{\acute{y}}{\acute{z}} + y\big(k_1 r^2 + k_2 r^4 + k_3 r^6\big) + 2p_1 xy + p_2\big(r^2 + 2y^2\big)
\end{aligned}\ ,
\tag{1}
$$

where:

- $f$—focal length;
- $x_0, y_0$—principal point position;
- $k_1, k_2,\ k_3, p_1, p_2$—additional parameters;
- $\acute{x},\ \acute{y}, \acute{z}$—coordinates of object point (after rotation and shift to camera frame);
- $r$—point radius in the image plane.

Many methods of camera calibration have been proposed in the literature on photogrammetry and computer vision [4,8–12]. In general, two groups here can be distinguished: Methods based on purposefully created calibration test fields, and self-calibration methods (on-the-job calibrations) that utilize images of regular scenes [1,6,9,13].

Calibration test field methods can be divided on the basis of the location of the calibration test fields, in and outside the laboratory. Both 2-D and 3-D fields can be utilized in laboratories, while outdoors is mostly 3-D. Calibration methods can also differ in the models describing the object–image relationship. In computer vision, mostly the homographic function is utilized, while photogrammetry uses collinearity equations [1,6]. A number of calibration test fields have been described in the literature, including 2-D [11,12,14] and 3-D [1,6,12,15] versions. An often-implemented kind of calibration test field is the checkerboard pattern [13]. Popularized through its inclusion in multiple software packages, such as OpenCV [16], Agisoft Metashape [17], and MATLAB [18], the checkerboard is a convenient way to calibrate a camera. However, it is not useful for large object–camera distances [19]. Checkerboard patterns need to be either projected or printed on a flat surface, such as a computer screen or paper. In either case, the size of the field is limited. Considering the calibration of a camera focused on long distances, it is unfeasible to satisfy the above conditions using the checkerboard field.

Leading manufacturers of large format photogrammetric cameras have their own proprietary laboratory calibration methods. Z/I Imaging utilizes a multiple-collimators device, which projects a dense grid of reference points on the sensor frame [20]. Vexcel Imaging implements a 3-D calibration rig sized $12 \times 2.4 \times 2.5$ m$^3$ [21]. The calibration of such cameras is usually highly stable and requires updating only once every several years. In-between calibrations manufacturers recommend monitoring IO stability by checking outside camera calibration test field image bundle adjustment results. Upon acquiring this type of camera, the client is provided with a camera metric with all the IO and AP parameters listed. Owing to its high cost and limited access, the procedure is not applicable to consumer-grade cameras with unstable IO elements. The instability of the IO necessitates repeated regular calibration procedures.

On-the-job calibration is rapidly gaining popularity, especially owing to a rise in UAV photogrammetric surveys [22]. It is most convenient for the end-user if camera calibration is performed within the bundle adjustment of the survey images. The process, simply put, adds unknowns and equations to bundle adjustment, and, in the end, provides information about distortion and the IO. Although it appears to be the perfect solution, it has problems that are challenging for inexperienced users to identify. The correlation between the distance to the object and focal length is a primary one. For flat-terrain images, the EO (exterior orientation) and IO are highly correlated and can lead

to gross errors in 3-D reconstruction [1]. Other parameters can be used to compensate for errors within a bundle without appearing in solutions as standard errors [23]. This may lead to incorrect reconstruction—something that is often not considered by end-users. Accuracy can be increased by including oblique imagery [22–24]. Such flight planning is a heavy burden for APM.

Implementation of on-the-job calibration in APM is inefficient and can be burdened with hard-to-detect geometric errors. To achieve the required accuracy, the utilized camera should be qualified as 'metric'. A compromise can be found by using medium format cameras. They can be implemented in production the same way as large format cameras, with only a small loss of accuracy [25]. Experiences in close-range photogrammetry prove that industrial cameras are becoming more stable [26]. Manufacturers provide camera metric as well as calibration services.

Though the stability of medium format cameras is increasing, it is recommended that camera calibration be carried out separately from the capturing of the survey images for highly accurate photogrammetric missions. Calibration should be performed regularly as well as every time a repair (ex. shutter replacement) is carried out. This would require a calibration method that uses a local test field—a group of points whose positions are known to a high degree of accuracy. Photogrammetric camera calibration using a test field is similar to self-calibration. It is a bundle adjustment with additional unknown parameters. However, in this particular case, the images used depict only the test field, and the only points used in the process are the known control points.

A few rules need to be followed for successful calibration [6]:

- The desired network should include various angles, positions, and points of view.
- There should be a variation in the scale of the images. In the case of a 2-D calibration test field, this means including multiple side-views.
- The comprehensive and unsystematic coverage of camera frames with points is required due to distortions' poor extrapolation properties.
- A large number of images and control points are needed for high observational redundancy.

The calibration process RE of control points is used to measure quality. In close-range photogrammetry, RE is expected to be smaller than 0.1 pixel [6]. A similar level is achieved in large format camera calibration [20,21]. Outdoors, a calibration value of 0.5 pixel is expected but with the inclusion of the GPS-INS system [20,21]. For medium format cameras, such standards have yet to be established. Though, if the accuracy of the end product is supposed to reach 1 pixel, the RE of camera calibration should not exceed 0.2-0.3 pixel.

The calibration process, in photogrammetry, utilizes bundle adjustment to establish IO and AP [1,4,6,27]. It relies on an adjustment process, where all equations describing the relationship in between camera and calibration test field markers are solved simultaneously. The equations include unknowns of cameras IO and AP as well as EO of all images (as in Eq. 1).

There are many commercial and free software calibration implementations that utilize 2-D and 3-D calibration fields. In computer vision, OpenCV is a well-established standard. It provides a transparent and flexible API (application programming interface), covering a large set of distortion coefficients, additional parameters, and supporting various camera models (pinhole, fisheye, omnidirectional). OpenCV calibration is based on the Zhang solution [9], though current implementation is capable of 3-D handling and utilizes global Levenberg–Marquardt optimization. Being very flexible in choosing the mathematical model of calibration, OpenCV implementation provides limited tools to handle input data and deal with errors:

- No a priori accuracy of control points can be provided by the user.
- No a priori accuracy of image measurements can be provided as well.
- Tie points are not allowed.
- Using checkpoints is feasible but requires an additional portion of code.
- Correlation values for unknowns are unavailable to the user.

- There is no interface to handle loss function, so the user is left to deal with gross errors on his own.

Taking into account the above-mentioned limitations and accuracy demands, we decided to create a dedicated approach that guarantees more control over observations and provides rigorous error handling.

## 2. Methodology Overview

A comprehensive calibration methodology should include both a designated calibration test field as well as a data processing pipeline. A flowchart of our proposed method is presented in Figure 1.

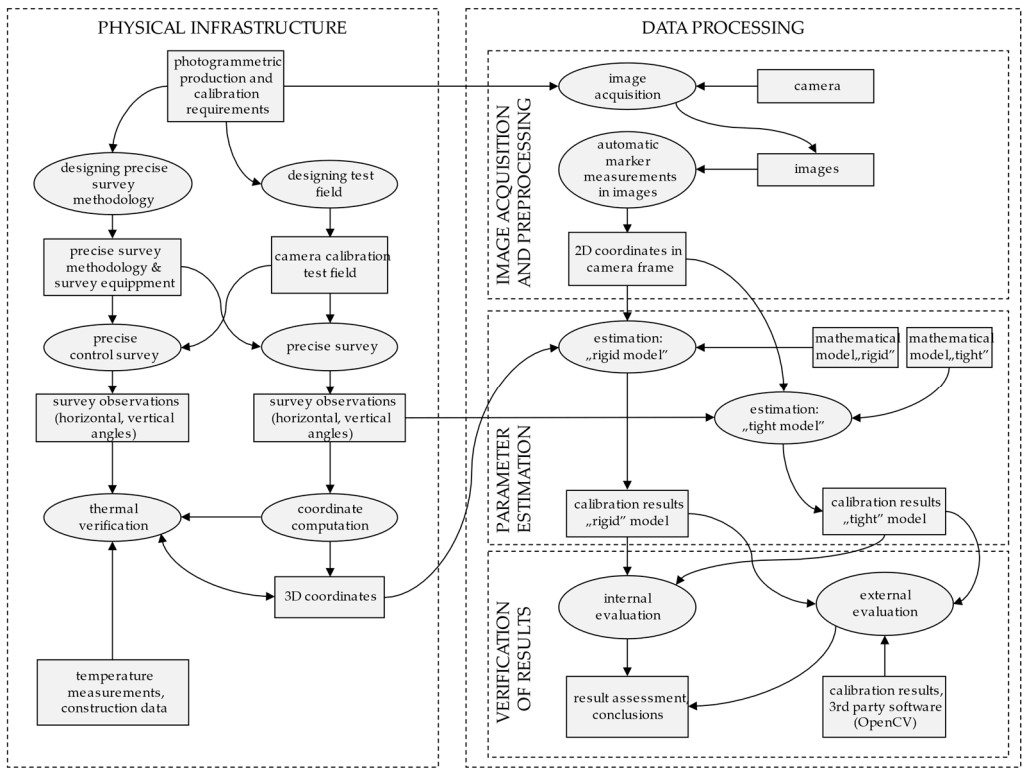

**Figure 1.** Flowchart representing the proposed methodology of camera calibration.

We start with a detailed description of the creation of the physical infrastructure, the calibration test field (Section 3). Laboratory calibration of medium format cameras requires a test field of larger dimensions compared to test fields typically used for calibration of close-range industrial imaging sensors. In such a case, one need not rely on repetitive distances (as in the checkerboard). To fulfill all calibration conditions appropriately, the field had to be of significant size, and include multiple high-accuracy control points that are easily automatically detectable in images. The purpose-built calibration test field was placed in the MGGP Aero company hangar. A precise survey was then needed to achieve submillimeter accuracy of positioning of the calibration test field markers. A conventional method of engineering surveying was performed in the established local coordinate system. In addition, the layout of the network points was strengthened with the use of invar scales. Since the calibration test field was placed on the hangar wall, an additional check was required to verify the markers' stability with changing thermal conditions. The survey was thus conducted twice, in the summer and in the winter, to check the differences in positioning.

Secondly, we continue with a comprehensive description of proposed data processing (Section 4). Two mathematical models for estimating camera calibration are proposed. The rigid model requires as an input 3-D coordinates of control points, while the tight model tightly integrates precise survey source

data and image measurements within the single adjustment process. Both models were implemented in the custom software.

Lastly, to validate the developed calibration infrastructure (physical and implemented adjustment approaches), the throughout study involving calibration of 3 cameras with a broad range of imaging characteristics was conducted. Each calibration was followed by an exhaustive examination of accuracy parameters for both models. In addition, for each camera, a comparative calibration using the OpenCV library was carried out. Details of the evaluation methodology as well as sensor characteristics are addressed in more detail in Section 5.

## 3. Physical Infrastructure

### 3.1. Design of Calibration Test Field

For productivity and flexibility of use, the calibration test field was located in a hangar where planes of the MGGP Aero company were stationed. The only location for the test field was the middle of the rear wall of the hangar, approximately 10 m long and 5.5 m high. Due to the construction of the hangar and its use, 3-D calibration test fields were unfeasible.

The calibration test field consisted of 238 markers distributed approximately every 30 cm, which covered an area 5 m wide and 4 m high (Figure 2). Six barcodes were placed on the wall to enable the automation of marker measurements on the images. Both markers and bar codes were UV printed on the 220 g/m$^2$ latex wallpaper, cut into single markers and affixed to the hangar wall by regular wallpaper glue.

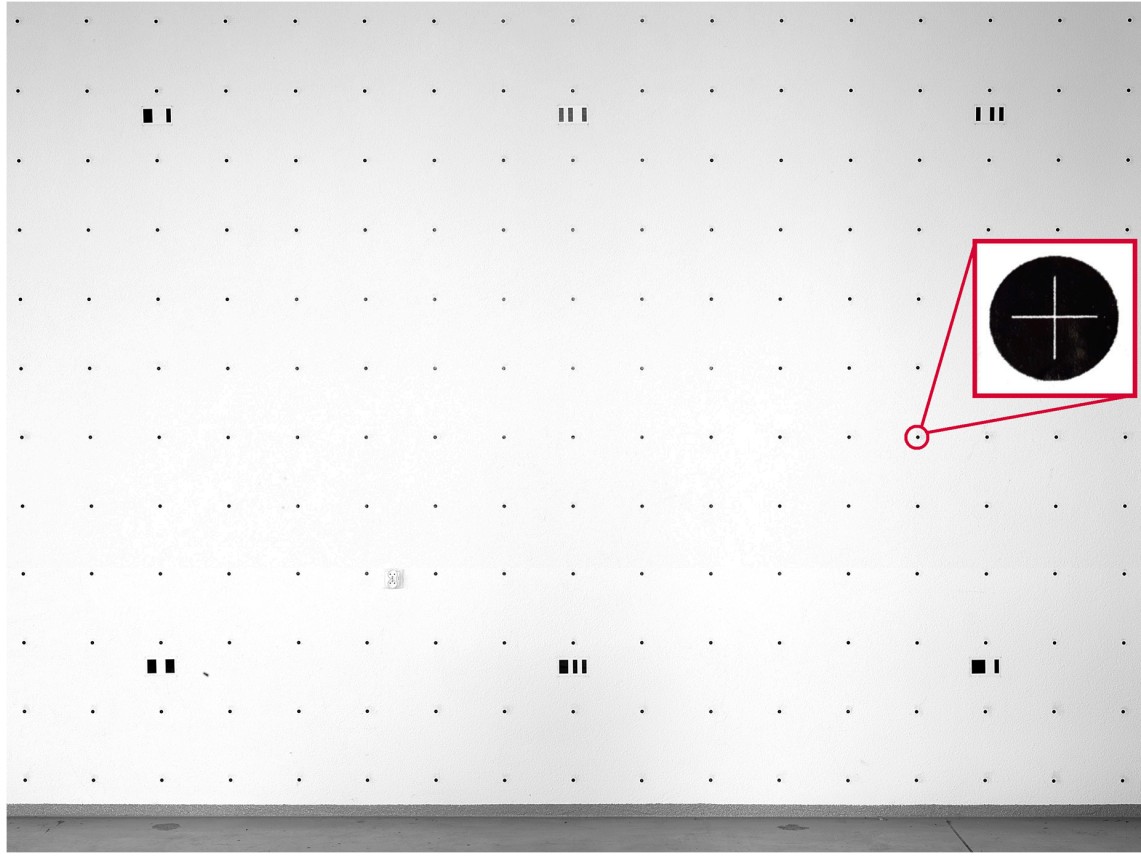

**Figure 2.** The calibration test field—a photograph of the network of markers and barcodes with one of the markers enlarged.

The markers were designed for precise targeting with a survey instrument. Assuming that the visual acuity of the observer's eye is 1 arc-min [28], the telescope magnification of a precise survey instrument is 30×, and the maximum survey distance is 15 m, and the cross-line cannot be thinner than 0.15 mm. Because the markers were observed at different angles, a 0.3-mm-thick cross-line was incorporated in their design (Figure 2).

### 3.2. Survey of the Calibration Test

In close-range photogrammetry, the 3-D control points are determined using the routine surveying method [29]. The accuracy-related requirements are often at the submillimeter level or higher. A control network with an accuracy of ± (0.05÷0.20) mm should be surveyed using the conventional method of engineering surveying with the additional use of a certain standard scale, such as the invar scale, leveling rods, or other equipment [29].

Coordinates of the field markers for the calibration were measured with an error not exceeding 0.3 mm. Figure 3 shows the design of the measurement network, together with error ellipses of the x and y coordinates for marginal points (P1, P2). They are located in the most unfavorable positions, i.e., at the border of the test field; therefore, their error ellipses will be the largest. Points B1 and B2 represent positions of the total station, which forms an observation base b.

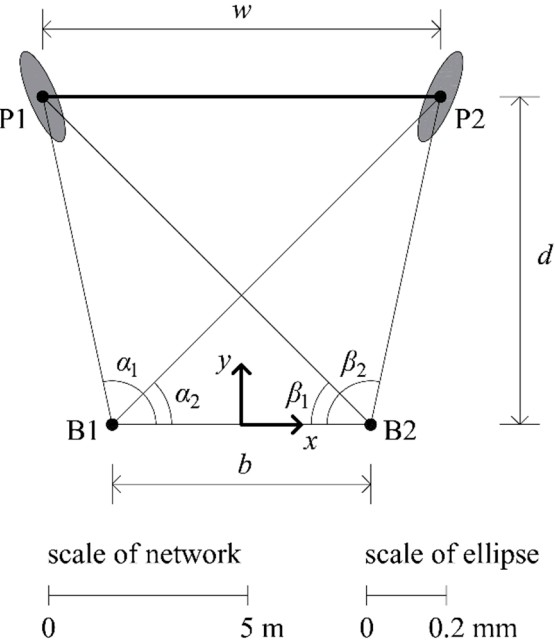

**Figure 3.** Design of the survey network and error ellipses of the critical points.

When designing the network, the following factors were taken into account: (a) The pre-assumed width of the test field w = 10 m, (b) $\alpha_2 = \beta_1 \geq 45°$ to ensure favorable angles for targeting, (c) minimization of the position error of marginal points P1 and P2 ($\sigma_{xy,P1} = \sigma_{xy,P2}$ = min), and the (d) even distribution of errors $\sigma_x$ and $\sigma_y$ in the measured marginal points ($0.5 \cdot \sigma_{y,i} \leq \sigma_{x,i} \leq 2 \cdot \sigma_{y,i}$; i = P1, P2)). Based on these conditions, the optimal dimensions of the network were determined as: b = 6.49 m, d = 8.25 m.

High-accuracy equipment was used for the measurements. To measure the angle, a precise total station Leica TCA2003 was used (0.5″ angle accuracy, 30× magnification) as well as auxiliary equipment: Precise EDM prisms with a centering accuracy of 0.3 mm and tribrachs to ensure a torsional rigidity of 1″. The accuracy of distance measurement using this instrument was 1 mm + 1 ppm, whereas it was 0.5 mm for a distance of 2÷120 m [30]. To scale the network, two horizontal invar subtense bars were used, and the error in their lengths was 0.03 mm (2 m Zeiss Bala bars were used). In addition, two 1.7-m invar leveling rods were used to verify the vertical length as determined by the survey measurement.

The group of control points (Figure 4) consisted of two observation stations (B1, B2) forming a base with an approximate length b, parallel to the plane of the calibration test field, and at a distance d from it. This group also consisted of two stable EDM prisms (11, 12), two horizontal subtense bars (81–84), two leveling rods (71–74), and six evenly spaced markers (nos. 0101, 0117, 0133, 1501, 1517, 1533).

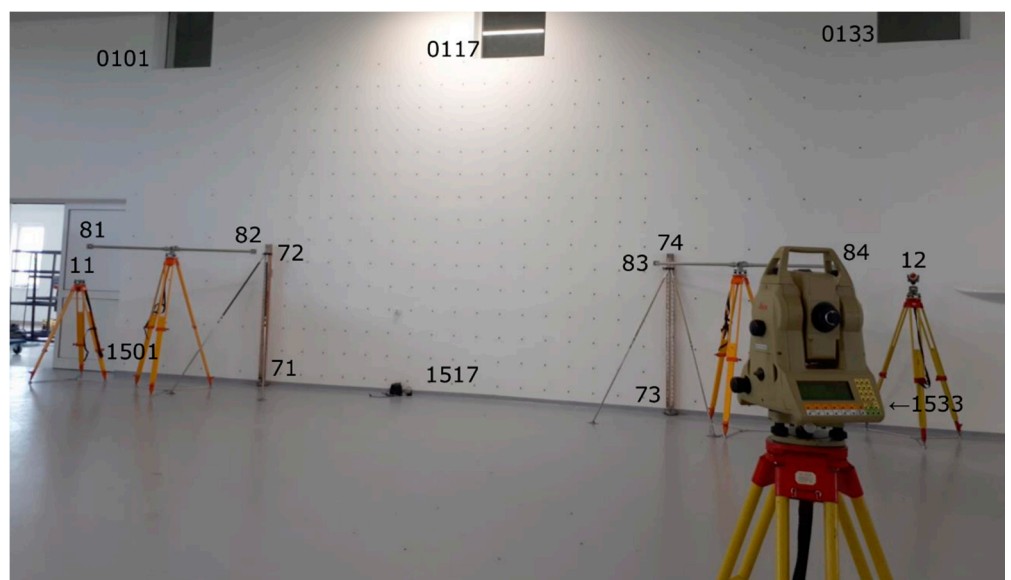

**Figure 4.** A view of the control points from station B2.

Precise angle and distance (if possible) measurements from stations B1 and B2 were carried out between the control points, while the positions of the markers were determined using the angle measurements. The visibility of all reference points and markers from both observation stations was required when setting all the network elements.

### 3.3. Evaluation of Measurement Accuracy

In the xy plane, the network was adjusted so that no deformation occurred (one fixed point, one fixed azimuth). The observational error for the angles was assumed to be 1″. The distance errors varied from 0.05 to 0.25 mm and were assumed to be 2.5 times the standard deviation of repeatedly measured distances. Values of lengths of 81–82 and 83–84 were adopted on the basis of the thermal expansion coefficient of invar, and their errors were assumed to be 0.03 mm. As a result of the network adjustment, the standard deviation obtained *a posteriori* was $\sigma_0 = 1.08$.

The heights of the control points were adjusted based on the slope distance to the slope and vertical distances between them. A very high consistency of observation was thus obtained. The height errors and the height differences errors did not exceed 0.05 mm after adjustment.

The xy coordinates of the markers were calculated using the angular intersection method. Furthermore, their heights were calculated based on vertical angles (measured) and horizontal distances from stations B1 and B2 (calculated earlier within the angular intersection). The heights were determined two times: From stations B1 and B2; therefore, it was possible to compare them. The obtained height differences were marked as dz. The largest difference was −0.30 mm and the mean absolute value deviation was 0.05 mm. The histogram of the differences is given in Figure 5a.

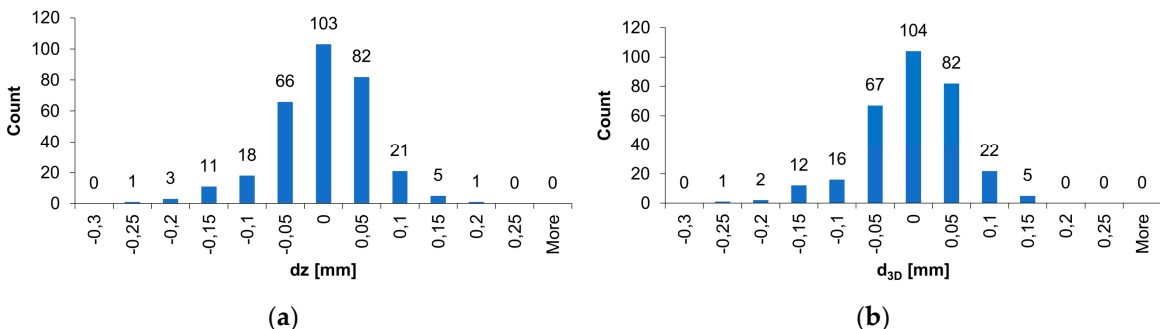

**Figure 5.** The histogram of: (**a**) differences in the heights of the markers, (**b**) shortest spatial distances between sight axes.

To verify the positions of the markers, an independent calculation was performed based on the shortest spatial distance between the lines created by the sight axes from stations B1 and B2. These straight lines needed to intersect at the center of the marker. However, a non-zero distance ($d_{3D}$) between them indicates an inaccurately determined marker position, resulting from the random error while targeting. Mathematically, the distance $d_{3D}$ can be determined as:

$$d_{3D} = \frac{V\left(\vec{B}, \vec{D_1}, \vec{D_2}\right)}{A\left(\vec{D_1}, \vec{D_2}\right)},\qquad(2)$$

where:

- $V$—the volume of the parallelepiped defined by three vectors—$\vec{B}, \vec{D_1}, \vec{D_2}$—that is given by the absolute value of the scalar triple product:

$$V = \left|\left(\vec{D_1} \times \vec{D_2}\right)\cdot\vec{B}\right| = \left\|\begin{bmatrix} dx_1 & dy_1 & dz_1 \\ dx_2 & dy_2 & dz_2 \\ bx & by & bz \end{bmatrix}\right\|,\qquad(3)$$

  taking into account these three vectors are not coplanar due to the measurement error.

- the area of the parallelogram defined by two vectors $\vec{D_1}$ and $\vec{D_2}$, which is given by the magnitude of the cross-product:

$$A = \left|\vec{D_1} \times \vec{D_2}\right| = \left\|\begin{bmatrix} i & j & k \\ dx_1 & dy_1 & dz_1 \\ dx_2 & dy_2 & dz_2 \end{bmatrix}\right\|,\qquad(4)$$

- $\vec{B} = [bx, by, bz]$—a vector representing the base, i.e., connecting the two observation stations B1 and B2; and

- $\vec{D_S} = [dx_s, dy_s, dz_s]$—a vector lying on the sight axis relative to position s (s = B1, B2), i.e., on the straight line connecting the station and the marker

The largest distance $d_{3D}$ was −0.29 mm and the mean absolute value of distances $d_{3D}$ was 0.05 mm (Figure 5b). Therefore, the results of the verification were highly consistent with the calculated marker heights using the angle intersection method. This indicates the correctness of the calculations and high measurement accuracy. For 95% of the markers, the differences in position determination in the two ways did not exceed 0.15 mm, and for 87% of the markers, they did not exceed 0.10 mm. The least accurately determined markers are shown in Figure 6, where their locations are expressed in the coordinate system of the test field (XYZ). Its origin was located at the center of gravity of all

markers while the X-axis coincided with the plane of regression of the test field fitted in all marker coordinates. The Z-axis coincided with the plumb line.

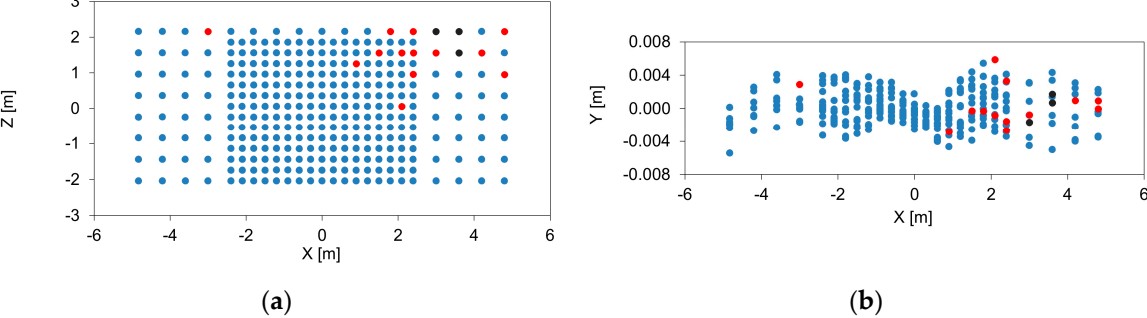

**Figure 6.** The locations of the markers as expressed in the coordinate system of the test field in: (**a**) the XZ plane, and (**b**) the XY plane. The red and black points indicate values of dz > 0.15 mm and dz > 0.20 mm, respectively.

Considering that errors in the estimated positions of the control points did not exceed 0.1 mm and the difference in dz were smaller than 0.2 mm for 98.7% of the markers, it can be assumed that the positional errors of the vast majority of the markers did not exceed 0.3 mm.

An additional control for measurement accuracy was used by comparing the lengths of the two extreme division markers on a vertical leveling rod, with the distance between them determined using survey measurement (points 71–72, 73–74). For the two rods, these differences were 0.18 and 0.07 mm, which confirmed the correctness of the work.

The least accurately determined markers (dz differences in the range from 0.15 to 0.3 mm) were obtained for points located in the upper-right part of the test field, probably owing to locally worse lighting conditions during the task. However, these values were consistent with the initial assumption of the accuracy of the markers.

### 3.4. Thermal Deformations of the Calibration Test Field

To determine the influence of ambient temperature on the stability of the positions of points in the test field, repeated measurements were performed when the temperature inside the hangar was 8 °C lower than at the first measurement. The survey was analogous to that described above, with not all points measured but only evenly distributed. This meant 17 odd columns, each with eight markers, for a total of 136 markers.

With two datasets (from primary and repeated measurements), it was possible to determine the impact of thermal changes on the stability of the markers' positions. Calculations were performed for a set of 136 points common to both measurements, and their coordinates were expressed in the same system.

The points were grouped in 17 columns to verify changes in the dX (horizontal) coordinates, and in eight rows to verify those in the dZ coordinates (vertical). In each of these groups, the coordinates and differences between them were averaged. These differences were calculated as: dX = X1 − X0 and dZ = Z1 − Z0 ("0" stands for the primary measurement and "1" represents repeated measurement).

The distribution of changes in dX and dZ depending on the location of the columns and rows justified the linear regression fitting: $dX = m_X \cdot X + b_X$, $dZ = m_Z \cdot Z + b_Z$ (Figure 7). In Table 1, the calculated parameters for linear regression are summarized. In addition, for the case of vertical deformation (Figure 7b), a set of dZ' values was considered, after reducing the set of dZ by the value for the lowest row (−2.1 m). This is due to the fact that this value appears to be an outlier, perhaps due to a different hangar wall structure near the floor, which may be subjected to different thermal deformation.

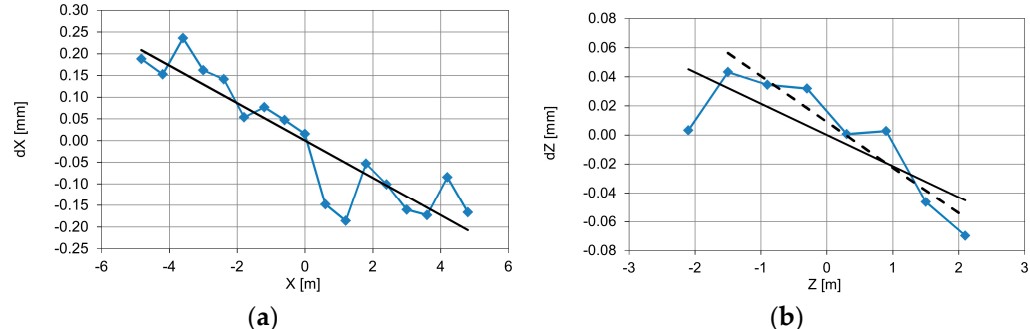

**Figure 7.** The distribution of changes in dX and dZ values depending on the location of the considered: (**a**) columns, (**b**) rows. The straight line means the fitted linear regression into the full set of results, while the dashed line was used for the dZ′ set.

**Table 1.** The statistics of a linear regression fitting into a set of values of dX and dZ (dZ′), and determining the thermal expansion coefficient $\alpha$.

| Parameter | dX | dZ | dZ′ | Unit |
|---|---|---|---|---|
| m | −0.043 | −0.022 | −0.032 | mm/m |
| $\sigma_m$ | ±0.015 | ±0.007 | ±0.005 | mm/m |
| df | 15 | 6 | 5 | |
| $R^2$ | 0.83 | 0.63 | 0.90 | |
| $\Delta T$ | | −8 | | K |
| $\alpha$ | 5.4 | 2.7 | 3.9 | ppm/K |
| $\sigma_\alpha$ | ±1.9 | ±0.8 | ±0.6 | ppm/K |

where: m—slope of the linear function; df—number of degrees of freedom; $R^2$—coefficient of determination; $\Delta T$—temperature change; and $\alpha$—thermal expansion coefficient.

The thermal expansion of the wall of the hangar was noticeable, and the coefficients of thermal expansion determined using measurements were 5.4 and 2.7 (or 3.9 without the outlier) ppm/K (along the X- and Z-axes, respectively). Although the wall is made of a homogeneous material, different thermal expansion coefficients were obtained for the X and Z axes. Perhaps the building structure itself causes additional loads (e.g., ceiling load in the vertical direction) that change the actual thermal expansion of the wall material. This may indicate that the deformation model is more complicated.

However, the change in the position of the markers was negligible. The mean absolute values of the differences between the measured and actual coordinates were 0.19 (X) and 0.06 mm (Z) when the temperature decreased by 8 °C. This means that the accuracy of the thermal change model was consistent with that of differences in the coordinates between successive epochs. Therefore, considering the thermal model when calculating the coordinates does not significantly improve the primarily determined coordinates.

## 4. Data Processing

Considering the limitations stated in Section 1.2, we decided not to rely on the OpenCV solution but implement a bundle adjustment solver with more flexibility in input data and error handling, as well as a rich set of accuracy measures to assess the results reliably. Additionally, we propose tightly integrating photogrammetric and survey observations: The ground control information is provided directly as geodetic surveys (horizontal/vertical angles), not as 3-D coordinates. The proposed approach allows for handling multiple surveys and gross error handling is done directly in the survey domain so individual angular or distance measurements can be suppressed by applying the loss function. Our tight approach takes over the adjustment of the survey, traditionally done independently from the bundle adjustment.

To calculate camera calibration (IO and AP), the solution to the optimization problem needed to be defined. Two models were investigated. The first, the rigid model, involved using the object coordinates

of the control points that were assigned no *a priori* errors. The second model, the tight model, did not directly use coordinates of the control points but integrated bundle and survey adjustment into one process. It tightly integrated the survey and photogrammetric measurements at the measurement level. Each model was addressed by implicitly defining the parameters as well as observations and their covariance. A cost function was defined and minimized.

### 4.1. Mathematical Model

The following mathematical notations were assumed to solve the optimization problem. The optimized vector of parameters augmented the following unknowns:

- $O = [O_{i=0}, \ldots, O_{i=I}]^T$ is the vector of coordinates of the center of projection of the images, with each vertical vector $O_i \in \mathcal{R}^3$.
- $\alpha = [\alpha_{i=0}, \ldots, \alpha_{i=I}]^T$ is the vector representing the orientation of *I* images, where each vertical vector $\alpha \in \mathcal{R}^3$ is composed of Euler angles and forms the rotation matrix $R(\alpha_i)$. The z-x-z (**α-ν-κ**) rotation sequence [27] was used as it properly conditions bundles with an inclination of cameras calling in −45° to + 45° range.
- $c = [x_0, y_0, f]^T$ represents IO parameters of a pinhole camera, a principal point coordinates ($x_0$, $y_0$), and a focal length (f) of the calibrated camera; thus, $c \in \mathcal{R}^3$.
- $e = [k_1, k_2, k_3, p_1, p_2]^T$ represents the vector of coefficients of camera distortion with up to five elements, three radial (k) + two tangential (p) (decentering) coefficients.
- $P = [P_{j=0}, \ldots, P_{i=J}]^T$ is the vector of *J* control points, with each vector $P_j \in \mathcal{R}^3$.

The parameters were constrained via the observed quantities, each of which is denoted by *obs*. The following observations and their a priori standard deviations were used depending on the optimization model used:

- $\left(p_{ij}^{obs}, \sigma_{ij}^p\right) \in (\mathcal{R}^2, \mathcal{R}^2)$ is the image measurements of the *j*-th control point to *i*-th image:

$$f_1 : p_{ij}^{obs} = \begin{bmatrix} x_0 - f\frac{\acute{x}}{\acute{z}} + \Delta x(e) \\ y_0 - f\frac{\acute{y}}{\acute{z}} + \Delta y(e) \end{bmatrix}, \tag{5}$$

  where:
- $\acute{x}, \acute{y}, \acute{z}$ are elements of a homogenous vector: $\overline{p}_{ij} = \mathbf{R}(\alpha_i)(P_j - O_i)$ and $\Delta x(e), \Delta y(e)$ are corrections for distortion.
- $\left(\beta_{jkl}^{obs}, \sigma_{jkl}^\beta\right) \in (\mathcal{R}, \mathcal{R})$ represents tachymetric measurements of the horizontal angles to the *j*-th control point from station *k* with reference (tie) to station *l*:

$$f_3 : \beta_{jkl}^{obs} = atan\frac{Y_k - Y_j}{X_k - X_j} - atan\frac{Y_k - Y_l}{X_k - X_l}. \tag{6}$$

- $\left(\gamma_{jk}^{obs}, \sigma_{jk}^\gamma\right) \in (\mathcal{R}, \mathcal{R})$ are tachymetric measurements of the vertical angles to *j*-th control point from station *k*:

$$f_4 : \gamma_{jk}^{obs} = acos\left(\frac{Z_j - Z_k}{d(P_j, S)}\right), \tag{7}$$

where: *d* is the slant distance, and *k* and *l* correspond either to station B1 or B2 ($k, l \in \{B1, B2\}$) as shown in Figure 3, whose coordinates were determined by geodetic measurement and free adjustment of the geodetic network.

By assuming that each group of observations forms a vector, the corresponding vectors of weights *w*, as well as corresponding corrections *v* subjected to minimization within the solution to the optimization problem, can be explicitly formulated:

- $w^p = \left[ \sigma^p_{i=0,j=0}{}^{-2}, \ldots, \sigma^p_{i=I,j=J}{}^{-2} \right]^T$ and $v^p = \left[ v^p_{i=0,j=0}, \ldots, v^p_{i=I,j=J} \right]^T$,

- $w^\beta = \left[ \sigma^\beta_{j=0;k,l\epsilon\{B1,B2\}}{}^{-2}, \ldots, \sigma^\beta_{j=J,k,l\epsilon\{B1,B2\}}{}^{-2} \right]^T$ and $v^\beta = \left[ v^\beta_{j=0;k,l\epsilon\{B1,B2\}}, \ldots, v^\beta_{j=J,k,l\epsilon\{B1,B2\}} \right]^T$,

- $w^\gamma = \left[ \sigma^\gamma_{j=0;k\epsilon\{B1,B2\}}{}^{-2}, \ldots, \sigma^\gamma_{j=J;k\epsilon\{B1,B2\}}{}^{-2} \right]^T$ and $v^\gamma = \left[ v^\gamma_{j=0;k\epsilon\{B1,B2\}}, \ldots, v^\gamma_{j=J;k\epsilon\{B1,B2\}} \right]^T$,

where $\sigma$ is the a priori standard deviation of observation.

The cost function (L) was minimized following the least-squares approach:

$$L(v,w) = v^T diag(w) v = \begin{bmatrix} v^p \\ v^\beta \\ v^\gamma \end{bmatrix}^T \begin{bmatrix} diag(w^p) & & \\ & diag(w^\beta) & \\ & & diag(w^\gamma) \end{bmatrix} \begin{bmatrix} v^p \\ v^\beta \\ v^\gamma \end{bmatrix} = min, \qquad (8)$$

where $diag(.)$ stands a diagonal matrix with elements of a vector on a diagonal.

Table 2 presents a summary of the implemented models.

**Table 2.** Models implemented for bundle adjustment with camera calibration.

| | Model | |
|---|---|---|
| | **Rigid**—Coordinates of control points could not be adjusted; they were kept fixed, which rendered the network more rigid. | **Tight**—This model provided the tight integration of photogrammetric and survey observations; it directly used survey angular observations. |
| **Components** | | |
| **Parameters** | $x = \left[ O^T, \alpha^T, c^T, d^T \right]^T$ | $x = \left[ O^T, \alpha^T, P^T, c^T, d^T \right]^T$ |
| **Types of equations** | $p^{obs}_{ij} + v^p_{ij}$ $= p\left( O_i, \alpha_i, c, d_n, P_j = const \right)$ | $p^{obs}_{ij} + v^p_{ij} = p\left( O_i, \alpha_i, c, e_n, P_j \right)$ $\beta^{obs}_{jkl} + v^\beta_{jkl} = \beta\left( P_j \right)$ $\gamma^{obs}_{jk} + v^l_{jk} = \gamma\left( P_j \right)$ |

## 4.2. Software Implementation

To facilitate calibration calculation for the created test field, the calibration software Xtrel was implemented in C++ [31]. It uses the Ceres Solver library [32] to solve nonlinear problems. Xtrel can handle many cameras in one adjustment process and can deal in a single adjustment process with photogrammetric and survey angle measurements (tight approach). The analytical derivatives of Equation (5), and numerical derivatives of Equations (6) and (7) were used. The interface to apply the loss functions (the one available in Ceres) was provided and was used in all experiments. The Xtrel solver provided detailed reports, including such accuracy measures as standard deviation and mean error. Moreover, the option of fixing any IO and AP parameters within the process was provided. Three kinds of points were allowed in the process, tie points, control points, and check points, where this provided the ability to use versatile calibration test fields and perform a multitude of experiments. Rotation can be represented using Euler angles (z-x-z, or x-y-z sequence). The Xtrel solver was provided with sufficient approximations of the unknowns before commencing with the adjustment.

## 5. Experimental Results

### 5.1. Experiment Design

A series of experiments were designed and conducted to evaluate the physical calibration infrastructure and mathematical models developed in this study. To make our experiments representative, we decided to test the calibration methodology for sensors inducing a broad range of imaging geometry. We performed calibration for 3 cameras: 2 medium format cameras, each with a lens of different focal length (narrow and normal angle), and one industrial camera (wide angle).

The evaluation of calibration results was two-fold. Firstly, a detailed, "internal", statistical analysis of accuracy parameters was carried out for each camera individually. This step involved examining such parameters as:

- RE;
- Root mean square errors (RMSE) for check points;
- Standard deviation of parameters; and
- Correlation coefficients.

Secondly, the "external" evaluation was conducted. This step involved:

- Comparison of results between examined models (rigid and tight); and
- Comparing parameters obtained using proposed models with those obtained using 3rd party software (OpenCV library).

### 5.2. Involved Sensors and Data Acquisition

Three cameras were calibrated to test the proposed method: the medium format camera Phase One iXM, medium format camera Phase One iXU, and small format, industrial camera Prosilica GT3400C (Table 3). There were 238 markers within the calibration test field. Out of all points recorded on the image, 10% were set aside as check points. As indicated in Section 4, each of 3 cameras was calibrated 3 times (tight model, rigid model, OpenCV), resulting in 9 calibrations in total. Figure 8 shows the exemplary geometry of acquisition, which was similar for all sensors. Nine images were taken for each camera, so that each frame was maximally filled with markers while maintaining maximum diversity in the orientation in between images. The biggest versatility in angles was achieved for the Prosilica camera, due to its short focal length.

**Table 3.** Parameters of the calibrated sensors.

|  | Camera | | |
|---|---|---|---|
|  | **Phase One iXU-RS1000 100 MP** | **Phase One iXM 100 MP** | **Prosilica GT3400C** |
| **Resolution [px]** | $11{,}608 \times 8708$ | $11{,}664 \times 8750$ | $3384 \times 2704$ |
| **Detector size [µm]** | 4.60 | 3.76 | 3.69 |
| **Nominal focal length [mm]** | 70 | 40 | 6 |

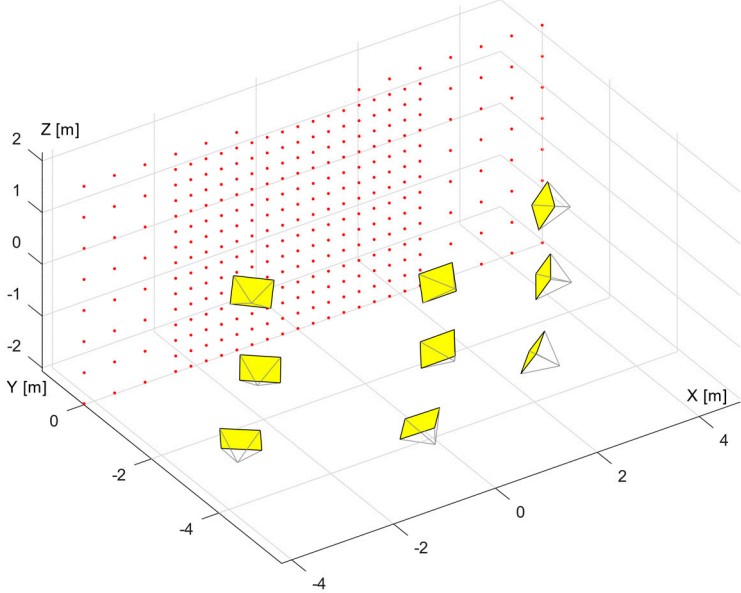

**Figure 8.** Exemplary imaging geometry (Phase One iXM): camera thumbnails and control points.

*5.3. Results*

For each camera, we estimated a full set of calibration parameters using OpenCV solution and our rigid and tight solutions. Table 4 summarizes the IO parameters for the three compared methods. Figures 9 and 10 show the distribution of radial and tangential distortions for the tight approach. Resultant distortion for the other two approaches is highly similar. IO parameters are highly similar except for the Prosilica (6 mm) camera, where OpenCV results give 6 pixels shorter focal length than for rigid and tight approaches. Remarkably, the Phase One iXU (70 mm) camera presents lower than pixel-level radial distortion in contrast to the Prosilica (6 mm) camera, where distortions reach 160 pixels (Figure 9). Decentering distortion is largest for Phase One iXM (40 mm) and reaches up to 4 pixels, while for other cameras it does not exceed 1 pixel (Figure 10).

Table 5 provides the RE for control points and RMSE for check points. RE for OpenCV and rigid solutions are almost identical, while for the tight solution it is consistently smaller. The clear almost double difference in favor of this solution applies to both Phase One cameras. The RMSE of the depth component of the check points (Y coordinate, Figure 8) was always significantly larger than components of the object plane. It is expected for nearly every photogrammetric network. RMSE for both Phase One cameras are similar between all models. For the Prosilica camera, the smallest RMSE was achieved with the tight model and the biggest RMSE was achieved with OpenCV.

In calibration networks, some parameters of the camera can be strongly correlated. Moreover, the parameters of the internal orientation were correlated with those of the external orientation. We thus calculated and showed the correlation coefficients (IO-AP and IO-EO).

Table 6 summarizes the correlation coefficients for calibration parameters (tight approach). The correlations between the parameters of radial distortion were high, reaching at least the value of 0.9 in all cases, which is the immanent feature of the Brown model [4]. The highest correlation can be observed for k1, k2, and k3 coefficients. For all tested cameras, the values exceed 0.9. This does not significantly impact distortion modeling, as the final resultant radial shift is crucial. The correlation between the position of the principal point and decentering distortion was also large for both medium format cameras, exceeding the value of 0.8. Visibly smaller correlations were observed for the Prosilica camera, where the highest correlation reached 0.65. High values for this correlation can distort the principal point position, which is crucial for correct reconstruction of the bundle. For all cameras, the correlation of $f$ and radial distortion remains mostly well below 0.7.

Correlation between IO and EO was also analyzed, as both parameter sets are simultaneously solved in the calibration process. In Table 7, always the highest correlation coefficients for the principal point position and orientation angles $\alpha$, $\nu$ (azimuth, inclination) were presented. Similarly, to earlier described correlations for IO and AP, they are bigger for both Phase One cameras than for the Prosilica camera. This can be attributed here to the much more favorable acquisition geometry for a camera with a shorter focal length.

Due to high correlation (IO-AP, IO-EO), it was difficult to compare camera parameters by observing only how close the values of the estimates were. For example, the three radial distortion coefficients, when compared between calibrations, might have yielded significant differences, although they together resulted in nearly identical distortion vectors in each pixel. To provide additional comparisons of the results, for each camera, a grid of $11 \times 11$ object points was generated that was subsequently reprojected to the image frame, with the external orientation set to zero and the parameters obtained using the rigid and tight models. For each point in the grid, the discrepancies between the image coordinates of the models were calculated.

As shown in Table 8 and Figure 11, the discrepancies in image projections between the rigid and the tight approach were larger for medium format cameras than for the small format. The strong correlation for the Phase One iXU 70 mm camera exhibited discrepancies of magnitudes of up to two pixels between the rigid and the tight models (Figure 11) (a one-pixel discrepancy was produced by a difference in $x_0$). The rigid and tight solutions obtained for the Prosilica GT3400C were almost identical.

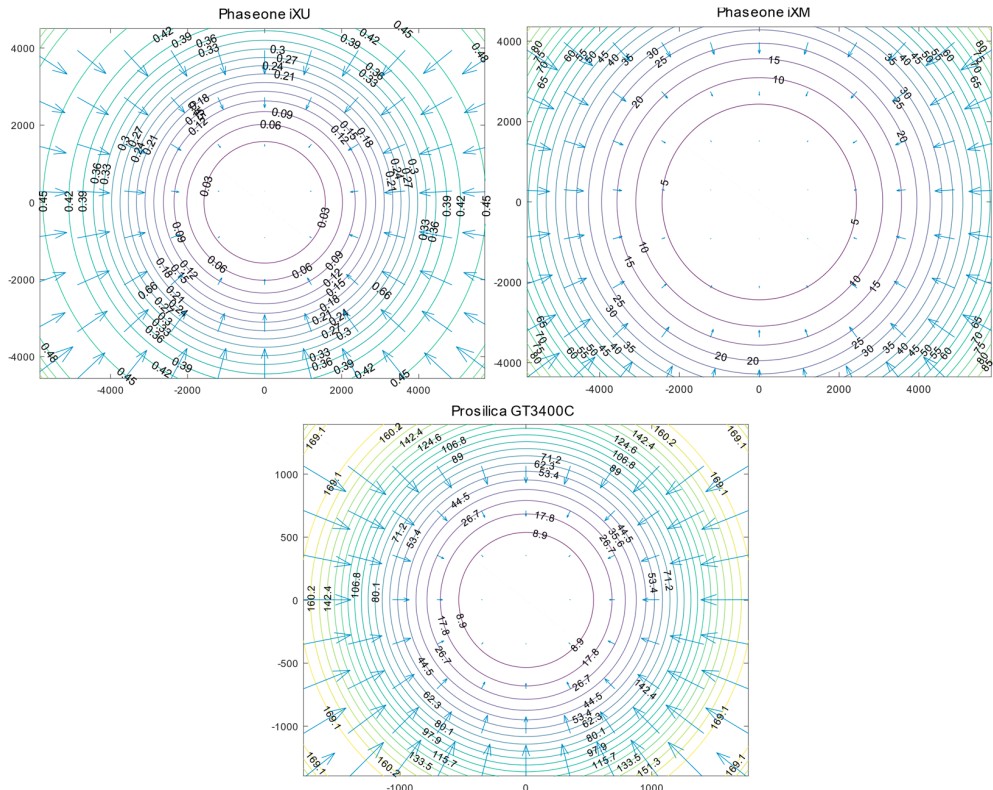

**Figure 9.** Distribution of radial distortion: magnitudes are shown as labeled contours and directions are indicated with arrows.

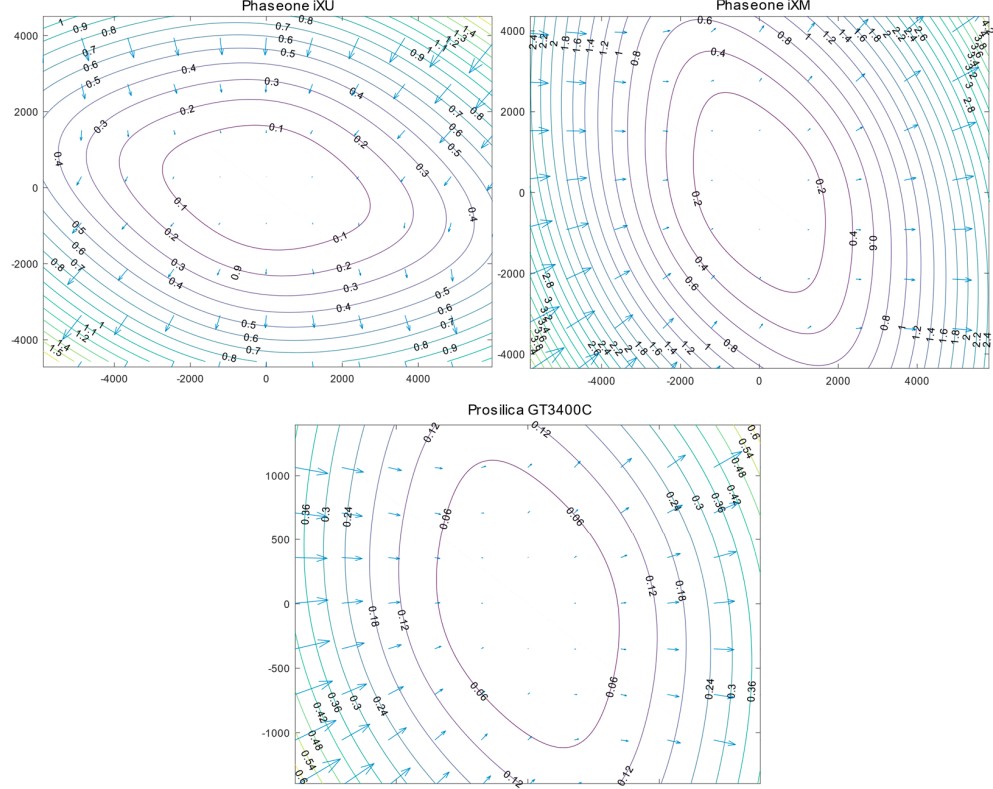

**Figure 10.** Distribution of tangential distortion: magnitudes are shown as labeled contours and directions are indicated with arrows.

**Table 4.** Resulting IO of calibration using the implemented models. Units in [px]. * values unavailable in OpenCV.

| | | Phase One iXU-RS1000 100 MP, 70 mm | | | Phase One iXM 100 MP, 40 mm | | | Prosilica GT3400C, 6 mm | | |
|---|---|---|---|---|---|---|---|---|---|---|
| | | OpenCV | Rigid | Tight | OpenCV | Rigid | Tight | OpenCV | Rigid | Tight |
| Camera parameters ±1σ | $f$ | 15,225.20 | 15,225.15 | 15,223.49 | 11,118.75 | 11,118.67 | 11,118.90 | 1679.88 | 1686.65 | 1686.58 |
| | | * | ±0.67 | ±0.55 | * | ±0.27 | ±0.23 | * | ±0.20 | ±0.18 |
| | $x_0$ | −30.47 | −30.42 | −29.53 | −59.08 | −59.31 | −58.54 | 6.07 | 6.08 | 6.05 |
| | | * | ±0.74 | ±0.83 | * | ±0.36 | ±0.37 | * | ±0.11 | ±0.10 |
| | $y_0$ | −10.07 | −8.93 | −8.45 | 10.84 | 10.84 | 11.70 | 40.14 | 40.30 | 40.41 |
| | | * | ±0.59 | ±0.71 | * | ±0.29 | ±0.30 | * | ±0.10 | ±0.11 |

* Values unavailable in OpenCV.

**Table 5.** RE for control points and RMSE for check points.

| | | Phase One iXU-RS1000 100 MP, 70 mm | | | Phase One iXM 100 MP, 40 mm | | | Prosilica GT3400C, 6 mm | | |
|---|---|---|---|---|---|---|---|---|---|---|
| | | OpenCV | rigid | tight | OpenCV | rigid | tight | OpenCV | rigid | tight |
| RE control points [px] | x | 0.39 | 0.39 | 0.16 | 0.37 | 0.37 | 0.16 | 0.34 | 0.38 | 0.30 |
| | y | 0.35 | 0.34 | 0.22 | 0.31 | 0.31 | 0.21 | 0.28 | 0.30 | 0.26 |
| RMSE check points [mm] | X | 0.11 | 0.11 | 0.13 | 0.11 | 0.10 | 0.12 | 0.38 | 0.18 | 0.08 |
| | Y | 0.25 | 0.22 | 0.19 | 0.22 | 0.22 | 0.23 | 0.56 | 0.24 | 0.19 |
| | Z | 0.09 | 0.07 | 0.06 | 0.06 | 0.06 | 0.07 | 0.46 | 0.12 | 0.06 |

Table 6. Pearson's correlation coefficients (k) for IO and APs.

| | Phase One iXU-RS1000 100 MP, 70 mm | | | | | | | | Phase One iXM 100 MP, 40 mm | | | | | | | | Prosilica GT3400C, 6 mm | | | | | | | |
|---|---|---|---|---|---|---|---|---|---|---|---|---|---|---|---|---|---|---|---|---|---|---|---|---|
| | f | $x_0$ | $y_0$ | $k_1$ | $k_2$ | $k_3$ | $p_1$ | $p_2$ | f | $x_0$ | $y_0$ | $k_1$ | $k_2$ | $k_3$ | $p_1$ | $p_2$ | f | $x_0$ | $y_0$ | $k_1$ | $k_2$ | $k_3$ | $p_1$ | $p_2$ |
| f | 1 | 0.00 | −0.02 | **−0.44** | 0.31 | −0.25 | −0.04 | −0.04 | 1 | 0.02 | 0.01 | **−0.69** | **0.57** | **−0.49** | 0.00 | 0.00 | 1 | −0.11 | 0.30 | **−0.65** | **0.56** | **−0.50** | −0.03 | −0.03 |
| $x_0$ | | 1 | 0.01 | 0.00 | 0.00 | 0.00 | **0.92** | 0.00 | | 1 | −0.01 | −0.01 | 0.00 | 0.00 | **0.90** | −0.02 | | 1 | −0.13 | 0.00 | 0.00 | 0.00 | **0.65** | −0.06 |
| $y_0$ | | | 1 | 0.01 | −0.01 | 0.01 | 0.02 | **0.85** | | | 1 | 0.01 | −0.01 | 0.01 | −0.01 | **0.81** | | | 1 | −0.01 | 0.00 | 0.00 | −0.10 | **0.46** |
| $k_1$ | | | | 1 | **−0.96** | **0.90** | 0.00 | 0.01 | | | | 1 | **−0.97** | **0.91** | 0.00 | 0.00 | | | | 1 | **−0.97** | **0.92** | −0.03 | −0.02 |
| $k_2$ | | | | | 1 | **−0.98** | 0.00 | −0.01 | | | | | 1 | **−0.98** | 0.00 | −0.01 | | | | | 1 | **−0.99** | 0.03 | 0.00 |
| $k_3$ | | | | | | 1 | 0.00 | 0.01 | | | | | | 1 | 0.00 | 0.01 | | | | | | 1 | −0.03 | 0.00 |
| $p_1$ | | | | | | | 1 | 0.01 | | | | | | | 1 | −0.01 | | | | | | | 1 | −0.04 |
| $p_2$ | | | | | | | | 1 | | | | | | | | 1 | | | | | | | | 1 |

* Values of |k| > 0.4 are presented in bold. Values of |k| > 0.8 are underlined.

**Table 7.** Pearson's correlation coefficients (k) between principal point and image orientation (angles)–ranges of values.

|  | Phase One iXU-RS1000 100 MP, 70 mm | Phase One iXM 100 MP, 40 mm | Prosilica GT3400C, 6 mm |
|---|---|---|---|
| $x_0$ vs. $\alpha$ angle | [−0.90, −0.88] | [−0.88, −0.70] | [−0.77, −0.60] |
| $y_0$ vs. $\nu$ angle | [+0.75, +0.79] | [+0.71, +0.90] | [+0.57, +0.73] |

**Table 8.** Differences in projections between rigid and tight solutions.

|  | Phase One iXU-RS1000 100 MP, 70 mm | Phase One iXM 100 MP, 40 mm | Prosilica GT3400C, 6 mm |
|---|---|---|---|
| $\Delta xy_{max}$ [px] | 2.30 | 2.22 | 0.25 |
| $\Delta xy_{mean}$ [px] | 1.17 | 1.35 | 0.13 |

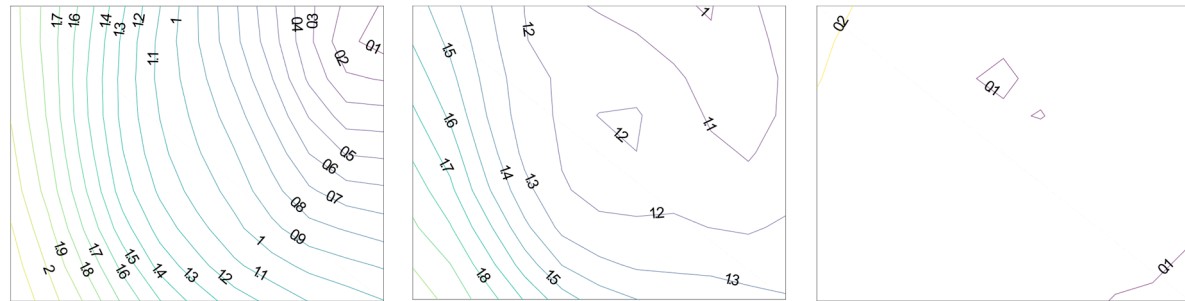

**Figure 11.** Absolute differences in projections (Euclidean distance) between rigid and tight solutions. From left: Phase One iXU-RS1000 (70 mm), Phase One iXM (40 mm), and Prosilica GT3400C (6 mm).

## 6. Conclusions

This paper proposed a method for camera calibration consisting of a calibration test field and the relevant software. The infrastructure is dedicated to the calibration of many types of cameras, particularly medium format cameras used in APM. The proposed model of calibration, referred to as the tight approach, integrates bundle adjustment and survey adjustment in a common process. Coordinates of control points are utilized by the solver neither as fixed parameters nor as observations but are treated as parameters of functions of surveyed angles and measured image coordinates. The proposed approach ensures that potential gross errors are suppressed by a loss function right at their sources: survey and photogrammetric measurements, preventing brute rejection of particular object point coordinates. Should each control point be measured from multiple survey stations, the gross error potentially affecting individual survey observation will be suppressed and will not affect the calibration results. In addition, the bundle adjustment solver developed here can be used to deal with on-the-job calibration (tie points can be included) or other typical bundle adjustment scenarios including adjusting terrestrial or airborne bundles (with or without camera calibration). In each use-case, a thorough accuracy analysis, including check points, was conducted.

The creation of a wall-sized calibration test field for camera calibration required the determination of markers' coordinates with the highest possible accuracy. The survey measurements ensured that the positional error was lower than 0.3 mm. It is worth highlighting that 3-D coordinates of test field markers were used for thermal analysis, while raw survey observations were implemented in the bundle adjustment. For small survey networks used in close-range photogrammetry, it is particularly important to scale the network with an invar pattern and perform precise angle measurements for the markers. In addition, high measurement precision enabled the determination of the thermal deformation of the test field based on observations carried out under different conditions. Nevertheless, no significant impact of thermal deformation on camera calibration was observed in this case.

Three cameras were used to test the proposed calibration infrastructure using three calibration approaches: rigid and tight solutions, as well as OpenCV implementation. In most cases, there were small differences in between the IO and AP parameters in the tested solutions. An unexpectedly large difference in focal length estimations was observed between the proposed solutions and OpenCV model for the Prosilica (6 mm) camera.

RE for the tight approach presented lower values in most cases. RMSE has the smallest values for the tight model in Prosilica camera calibration, while for Phase One cameras RMSE is similar for all implemented models. RMSE of check points clearly points to erroneous results in the OpenCV method for the Prosilica camera. The source of this error is unknown, but it has to be highlighted that the Prosilica (6 mm) camera has an ultra-wide-angle lens. This property could be further studied by comparing calibrations of different ultra-wide-angle lenses.

From the evaluation of IO-AP and IO-EO correlations, a conclusion has to be drawn that the values are a bit too high. The example of the Prosilica camera shows that the higher versatility of orientation angles substantially decreases those correlations. High correlations between the solved parameters diminish the reliability of the calibration process. Despite that, our proposed method is already utilized in photogrammetric survey for calibration of mainly medium format cameras. It should be added, however, that the industrial cameras are used in projects where expected RMSE for check points in aerotriangulation are set at the three ground sampling distance (GSD) level.

A drawback of our proposed infrastructure is its inability to avoid the particular correlation of some calibration parameters, especially medium format cameras. While creating the calibration test field, we were limited by the physical conditions set by the construction and use of the hangar. The only way of extending the depth of the calibration test field, without limiting the use space of the hangar, is to expand it using temporary bars with markers placed in front of the existing field. Since each time the calibration is performed the bars would be placed in different locations, their marker positions will not be constant in relation to the field control points. Thus, they will have to be treated as tie points, an option allowed in the proposed framework. This solution could reduce the correlation between IO-AP and IO-EO parameters and consequently lead to accuracy improvement.

**Author Contributions:** J.K.: Methodology, Software, Investigation, Writing the Original Draft. P.K.: Investigation, Writing the Original Draft, E.P.: Formal Analysis, Writing the Review, and Editing. K.P.: Supervision, Formal Analysis, M.S.: Investigation. All authors have read and agreed to the published version of the manuscript.

**Funding:** This research was founded from the agreement between MGGP Aero, Tarnow, Poland and AGH University of Science and Technology No. 5.5.150.531, and was assisted by AGH UST project Initiative for Excellence Research University.

**Conflicts of Interest:** The authors declare no conflict of interest.

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
