# Peer review of "Calibration of Industrial Cameras for Aerial Photogrammetric Mapping"

_remotesensing, doi:10.3390/rs12193130_

Round 1

Reviewer 1 Report

The paper presents a calibration method for use in photogrammetric survey production.

They have designed a  calibration test field. The field markers were measured with enough accuracy.

In the calibration process they have developed a analytical model of bundle adjustment.

Reviewer 2 Report

The authors present a methodology for the calibration of medium format photogrammetric cameras.

The literature review is satisfactory and covers most of the research field. Their methodology is well explained. In addition the setup an measurement of the calibration filed is very well explained. Finally the data acquisition procedure and the data processing are also very well explained and their conclusions are supported by their results.

I have the following questions:

The authors used the Brown calibration model, which does not include the affinity and shear component in the calibration parameters, can they comment about their choice.

What is the number of the control points used and what is the number of check point used?

Did they try to perform the calibration using a commercial software like agisoft or pix4d in order to compare their results with ones coming from a commercial software?

Finally, did they try to use a self-calibration approach and compare the results with the ones resulting from a pre calibrated camera?

Reviewer 3 Report

It is a comprehensive camera calibration study detailing from test field construction/measurement to the calibration software implementation. Its interest to the readers is high.

1) Although the presented work is very specific to the laboratory (test field) calibration of the medium format cameras, the title is too generic. I suggest to adapt the title to the content.

2) Writing English may need a medium level of editing preferably by a native English speaker or by an editorial office.

3) Line [73]. What is the reprojection error (RE) ? Is it the a posteriori precision of the image measurement at the control points?

4) Line [78, 79]. This sentence is too much confusing. Please, consider to re-write it.

5) The literature review is comprehensive and complete.

6) Figure 1. An enlarged view of a single target would be very informative. 

7) Line [203]. aiming --> targeting   ??

8) What kind of medium (material) is the target made of?
Paper? Cartoon? Plastic? ..?

9) Line [205]. It is not clear the statement in the parenthesis. A revision of the sentence is required. 

10) The text in between Lines [212- 230] should be extensively revised. Many grammatically and semantically incorrect sentences exist.

11) Line [234]. What does "critical points (P1, P2) " mean?
Are they the control points (markers) of the test field?

12) Line [236]. aiming --> targeting   ??

13) Line [237]. —σxy,P1 = σxy,P2 = min—
Put it in a separate equation line.
What does the "—" symbol mean here?

14) The same also applies to the equation piece in Line [238].

15) Figure 2. What are the points B1 and B2?
Are they the total station points?

16) The sentences in between Lines [271-276] should be edited. 

17) Lines [285, 286]: I do not understand how three coplanar vectors (B, D1 and D2) constitutes a parallelepiped.
Should it be a 3D triangle?
A figure depicting the vectors and shapes would be helpful.

18) Lines [373-380]. The rigid and the tight models are the two end points of the possible adjustment spectrum.
Why do not you use the following intermediate options?
- Free network adjustment?
- Control points with a priori precision values?

I believe that the current implementation is incomplete for a proper validation and justification.

19) Line [388]. z-x-z --> y-x-z   ??

20) Line [400].   the projection  -->  the image measurement   ??

21) Equation (8). The mathematical model of the tight version should be given in the open form.

22) Section 4. Unfortunately, some design problems apparently exist.
A) A perfectly planar control point distribution is not good to retrieve an accurate estimation of the focal length (or so called camera constant). The lack of depth geometry in the control network will deteriorate the depth related unknowns. A practical solution would be taking images at stations whose camera to the wall distances are significantly different. E.g. at Y=-2m, Y=-4m and Y=-6m, etc.

B) I would use the three intersecting walls (two vertical and one floor (or ceiling)) instead of one vertical one. Three intersecting walls would give a better geometry to be a control field.

C) In such a image acquisition geometry give in Figure 7, some of the internal orientation and external orientation parameters are highly correlated. In order to prevent this situation, some images are taken with 90 degree rotation (to left or to right sides).

D) Some unknown parameters would be in-significant, and some others may be in-determinable. An appropriate testing strategy should be embedded.

These issues must be clearly and objectively discussed in the text.

23) Table 6. It is obvious that k3 is highly correlated with k1 and k2. A posteriori precision of k3 should also be assessed here. Most probably, it should be excluded from the system. This version would improve the results.

Reviewer 4 Report

The paper deals with photogrammetry calibration. The present paper
presents many lacks mainly in term of reliability of the document and in
the organisation of the paper itself. While the introduction and the
description of the research domain is quite complete, even if some more
studies could be integrated in order to provide a more consistent
picture on the specific topic analysed, the methodological section is
too much confused and not well organized. The presentation of the method
should be organised with a first overall description, eventually
integrated with a graphical flowchart for going further with the
specific information regarding the theoretical framework and the
parameters involved and their fundamental correlations. Also the
experimental validation would need improvements in term of better
explaining the experimental setting.

Round 2

Reviewer 3 Report

All my comment and remarks have been replied by the authors. The manuscript can be considered for a possible publication in the RS journal.

Reviewer 4 Report

Authors have improved the scientific level of the paper